# Organic Food in the Diet of Residents of the Visegrad Group (V4) Countries—Reasons for and Barriers to Its Purchasing

**DOI:** 10.3390/nu13124351

**Published:** 2021-12-02

**Authors:** Andrzej Soroka, Anna Katarzyna Mazurek-Kusiak, Joanna Trafialek

**Affiliations:** 1Institute of Health Sciences, Siedlce University of Natural Sciences and Humanities, 2 Konarskiego Street, 08-110 Siedlce, Poland; wachmistrz_soroka@o2.pl; 2Department Tourism and Recreation, University of Life Sciences in Lublin, 15 Akademicka Street, 20-950 Lublin, Poland; 3Institute of Human Nutrition Sciences, Warsaw University of Life Sciences (WULS), 166 Nowoursynowska Street, 02-787 Warsaw, Poland; joanna_trafialek@sggw.edu.pl

**Keywords:** organic food, consumer behavior, the Visegrád Group, healthy lifestyle, nutritional values

## Abstract

This study aimed to determine the differences in the frequency of, reasons for, and barriers to purchasing organic food among the inhabitants of the Visegrád Group member states. The selection of the countries for the study was dictated by the fact that the countries of Central and Eastern Europe play the role of a niche market in the European organic food market. This research employed the method of a diagnostic survey and the discriminant function. A chi-squared test, ANOVA, and Fisher’s Post Hoc LSD test were also used to present differences in individual groups. This research shows that respondents from Poland, the Czech Republic, Hungary and Slovakia were guided by similar behaviors regarding the purchase of organic food. However, the attitudes of the respondents slightly differed between countries. In the case of the reasons for choosing organic food, the most important thing was that it is non-genetically modified food, especially for Polish consumers. The following were also mentioned: lack of chemical compounds (Slovaks and Czechs), high health value of such food (Czechs and Slovaks), and excellent taste (Hungarians). The most critical barriers against purchasing are the price (Poles and Hungarians), difficult access (Poles and Hungarians), and the short expiry time of such products (Slovaks).

## 1. Introduction

According to the American Center for Disease Control, 53% of human health depends on diet and lifestyle, 21% on the quality of the natural environment, 16% on genetic heritage, and 10% on the activities of health care [1,2].

According to WHO, about 70–80% of heavy metals such as cadmium, lead, mercury, but also pesticides and herbicides in the human body come from the food consumed. On an annual average, the average consumer introduces about 2.5 kg of agricultural and food chemicals into the body [2,3]. This is why introducing organic food to the daily menu is of great importance.

The food market is changing dynamically. New food production technologies are emerging, especially those with high and enriched nutritional value and functional food [4,5,6]. To a large extent, the changes concern the organic food sector, which contributes to maintaining good health due to maintaining metabolic processes in the body, which is an essential element of health prevention [7,8]. Organic food has a high nutritional value, ensuring safe consumption without harmful residues of fertilizers and preservatives, and excludes genetic modifications [7,9,10,11,12]. It impacts the increasing level of the search for organic food, little-processed food and food from certified organic farms by consumers from European countries [13].

The richest countries, i.e., in northern and western Europe, have the largest share in organic food sales. The impact on the level of consumption includes high household incomes and prices of substitute products. The leading countries are Denmark (13.3%), Sweden (10.5%), Switzerland (9.0%), Austria (8.6%), Luxembourg (7.3%) and Germany (5.1%). In postsocialist countries, organic food is also beginning to enjoy growing popularity, but its share in the consumption market is at a negligible level (less than 0.1%) [14,15], although there has been a systematic increase in residents’ awareness of the quality of consumed products and a healthy lifestyle [16,17,18]. The composition of food products, their origin, and production methods are subject to a much greater verification, focusing on ecological conditions [1].

In the V4 Group countries, the growing demand for ecological products is undoubtedly influenced by the introduced financial support for this type of manufacturing activity. The state of environmental awareness and the threats posed by the degradation of the natural environment on the quality of life also affect consumers’ motivation, which does not always translate into systematic pro-ecological purchases and sustainable practices [19]. There is a discrepancy between the attitudes and intentions of consumers and actual purchases [20,21], although there is a group of consumers for whom environmental awareness is crucial when shopping for food [22].

To eliminate these discrepancies, it is vital to learn the consumers’ reasons and barriers when purchasing organic food in countries where this problem is becoming increasingly important. This study aimed to determine the differences in the frequency, reasons, and barriers against buying organic food by residents of the Visegrád Group member states.

## 2. Materials and Methods

The research was conducted in 2016–2019 among the inhabitants of the Visegrád Group countries, which was established on 15 February 1991 as a regional form of cooperation between four Central European countries: Poland, the Czech Republic, Slovakia, and Hungary. These countries are linked both by neighborhood and similar geopolitical conditions and by a shared history, culture, and tradition [23]. The selection of the V4 Group countries was due to, within the European organic food market, the countries of Central and Eastern Europe have played the role of a niche market in it so far. Since 2004, the nature of this market has been influenced by determinants and regulations in force within the European Union. An additional reason for the selection of the V4 countries for this study was the fact that they had operated within the socialist system for 45 years, which imposed on their inhabitants a different way of thinking, culture, traditions, and models of consumer behavior than in western European countries [24].

Our research employed the method of a diagnostic survey with the use of the direct questionnaire technique. An original questionnaire was developed based on previous research on purchasing reasons and barriers [18,25,26]. The form consisted of two blocks of questions—the first concerned the frequency of, reasons for, and barriers against purchasing organic food. The second comprised questions on the characteristics of the respondents (nationality, age, sex, education, type of place of residence). The survey questionnaire is presented in Appendix A.

A five-point Likert scale was used to measure attitudes (1—Not important, 2—Low important, 3—Medium important, 4—Important, 5—Very important for the consumer), preceded by applying construction and validation procedures. The study conforms to the code of ethics of the World Medical Association and the standards for research recommendations of the Helsinki Declaration. To ensure confidentiality, all data were anonymized before analysis.

It was determined that the minimum sample size in each of the four countries surveyed was 738 respondents [27]. To calculate the sample size, the standard error of the estimate was taken as 0.03 and the confidence level as 0.95. The research applied the method of a diagnostic survey with the use of the direct questionnaire technique. The study was conducted using the Paper and Pencil Interview (PAPI) technique. An original questionnaire was developed. The questionnaire contained three research questions and additional questions defining the sociodemographic characteristics of the consumers. Participants were not paid for the study. The study was anonymous. Respondents gave verbal consent to participate in the study. The group of respondents was defined proportionally to the entire population of the surveyed countries, considering four age groups (20–34 years; 35–49 years; 50–64 years, 65 and more). In the next stage of selection, the number of women and men was proportionally determined. To facilitate the conducting the study, the established number of respondents was a sample representative of the four regions that occur in each of the countries: North-East—185; North-West—184, South-East—184, and South-West—185. Random selection was used, consisting of questioning considering the availability of respondents until the specified number of respondents in the groups was exhausted. The characteristics of the research sample are presented in Table 1.

The program Statistica 13.3 PL (StatSoft Inc., Krakow, Poland) was used for statistical calculations. The calculations were performed at the confidence level of 0.95, and the maximum error was set at 0.05. To determine which variables stood out in four naturally selected groups, the discriminant function analysis was used to study the differences between groups of objects based on a set of selected independent variables [28].

Discriminant function analysis carries out a multivariate test of differences between groups and approaches the problem by assuming that the conditional probability density functions $p(\vec{x}|y=0)$ and $p(\vec{x}|y=1)$ are normally distributed with mean parameters and covariance ((${\vec{\mu }}_{0},\sum_{0});(0,\sum_{0})$ and (${\vec{\mu }}_{1},\sum_{1}$) [29]. The logarithm of the probability ratios is more significant than a specific threshold T, so that:(1)$$(\vec{x}-{\vec{\mu }}_{0}{)}^{T}\overset{-1}{\underset{0}{\sum }}\left(\vec{x}-{\vec{\mu }}_{0}\right)+\ln|\underset{0}{\sum }|-{\left(\vec{x}-{\vec{\mu }}_{01}\right)}^{T}\overset{-1}{\underset{1}{\sum }}\left(\vec{x}-{\vec{\mu }}_{1}\right)-\ln|\underset{1}{\sum }|>T$$
where:

$w_{i}$—Regression coefficients,

${\vec{\mu }}_{k}$—Mean parameters,

$\sum_{k}$—Covariance.

Discriminant function analysis is used in correlation studies, i.e., when the causal relationships between variables are not well understood [28]. The research used a classification function to calculate the coefficients that were determined for each group of variables. Before the analysis, multivariate normality was analyzed, testing each variable for normal distribution. It was assumed that variable variance matrices are homogeneous in groups. Slight deviations were not that significant due to a large number of respondents in particular groups. Means for which the probability was less than *p* < 0.05 were considered statistically significantly different.

The chi-squared test, ANOVA, and Fisher’s Post Hoc LSD test were also used to present differences in individual groups.

## 3. Results

In the first stage of the research, differences in the frequency of purchasing organic food in the studied countries were identified. At *p* < 0.001, there were statistically significant differences in the frequency of organic food purchases between the inhabitants of individual countries. It was found that that Poles and Hungarians buy organic food very often, at 33.74% and 29.13%, respectively. On the other hand, the largest percentages of Slovaks and Czechs do so rarely, at 27.37% and 25.47%, respectively. (Table 2).

The highest average frequency of purchasing organic food on a 1–5 point scale was found in Poles $\bar{x}=3.48$, and to a lesser extent in Hungarians $\bar{x}=3.38$. It is followed by the Czechs $\bar{x}=3.14$ and the Slovaks $\bar{x}=2.99$. Fisher’s post hoc LSD test shows that there are significant differences in the frequency of purchases between Poland and the Czech Republic and Slovakia at the level of *p* < 0.001, between Hungary and the Czech Republic and Slovakia at the level of *p* < 0.001, and between Slovakia and the Czech Republic at the level of *p* = 0.026 (Figure 1).

In the second stage of the research, the respondents were asked about the reasons for purchasing organic food. The discriminant function model included 10 out of 11 reasons that were assessed. The model showed strong overall discriminatory power.

The highest discriminatory power was achieved among individual reasons by the lack of genetic modification of organic food. With *p* < 0.001, such a declaration was presented to the highest degree by Polish consumers and to a lesser extent by respondents from other countries. The classification function obtained almost two-times lower values in the reason of the lack of chemical compounds in organic food. At *p* < 0.001, the highest declarations were presented by consumers from Slovakia and the Czech Republic, while Polish respondents significantly reported the lowest. A critical reason, especially for Czech and Slovak consumers, was the high health value of organic food. The importance of the classification function in their case was significantly higher, with *p* < 0.001, compared to the Hungarian and Polish consumers. A vital reason, especially for Czech and Slovak consumers, was the high health value of organic food. A critical reason, especially among consumers from Hungary and the Czech Republic, and Slovakia, was the taste of organic food. In the case of Poland, the value of the classification function, with *p* < 0.001, was significantly the lower than consumers from other countries. A very similar situation was found for the nutritional value. The Slovaks presented the highest declarations, the Czechs and Hungarians to a slightly lower degree, and significantly the lowest, with *p* < 0.001 was the Poles. Furthermore, to the significantly lowest degree, consumers from Poland compared to other countries, with *p* < 0.001, paid attention to food processing and its calorific value. However, even with the reason of the lack of preservatives in organic food, the declarations of Polish consumers were significantly the highest, with *p* < 0.001. To a lesser extent, consumers from all countries indicated organic food as environmentally friendly and not causing allergies (Table 3).

In the third stage of the research, the respondents were asked about the barriers to purchasing organic food. All obstacles that were assessed were entered into the discriminant function model. Additionally, in this case, there was a solid overall discriminatory power in the created model. Compared to respondents from Slovakia and the Czech Republic, consumers from Poland and Hungary drew attention to the excessively high prices of organic food to the greatest extent, at *p* < 0.001. A very similar relationship, also at *p* < 0.001, was found related to the notion that organic food is difficult to obtain. The short-term consumption of organic food was indicated to the highest degree by consumers from Slovakia, while to a lesser extent by the Czech Republic and Hungary and to the lowest degree, with *p* < 0.001, by Polish respondents. Similar relationships occurred regarding the barrier related to the lack of trust in organic food. This barrier was indicated to the most minor extent by Polish consumers and to the greatest extent by Slovak respondents (Table 4).

## 4. Discussion

The goals set by the study were fully achieved. It was indicated that in the group of V4 countries, the frequency of consumption of organic products was at different levels. This was seen to the greatest extent by consumers from Poland and Hungary, and to a lesser extent by the other two countries. The study determined the frequency of consumption of organic products. Most of the available studies have focused on assessing the consumption of specific types of organic products, while in our study, we presented the problem in a general way. For example Spanish consumers mostly eat vegetables, tomatoes, fruits and eggs many times a week or every day (42.4%, 38.9%, 40.2% and 39.2% respectively) [30]. 

In comparison, jam and wine (8% and 6.2%, respectively, every day or many times a week) were the least consumed products [30]. The French very often (many times a week) consume organic fruit and vegetables, juices, meat, cold cuts (45%, 38%, 18.9%, 29%) [31], while German consumers consume organic meat and cold cuts several times a week (49%) [32]. In contrast, Danes are most likely to buy organic yogurt–48.6%, carrots–45.2%, oatmeal–43.1%, bananas–36.5%, and cooking oils–34.3%. On the other hand, in Switzerland the best-selling organic products are bananas (23.6%), tomatoes (17.1%), carrots (13.8%), apples (12.1%), lemons (10.8%) and peppers (10.1%) [1,17,33]. The demonstrated differences in the consumption of organic products in individual V4 countries may result from the actual differentiation or different interpretations of the definition of organic products and confusion with home and locally produced food. For example, Spanish consumers believe that all food made at home or bought in local markets is organic [30].

Previous studies about food choice drivers conducted in industrialized countries identified price [34], health [35] sensory appeal [36], convenience and ethical concerns [37,38] as the main reasons influencing consumers choice. These studies investigated reasons such as environmental, animal welfare, and local production, and they mainly focused on organic products [39,40] or specific/local food groups [41]. For example, a study conducted in Finland reported that health and ethical concerns were associated with higher consumption of fruit and vegetables and lower consumption of energy-dense foods [38]. Another study, conducted among young adults in the USA, reported that a positive attitude toward organically grown and local food was associated with higher consumption of fruit and vegetables and lower consumption of sugar-sweetened beverages [40]. In Asian countries, the main reasons for buying organic food are a concern for their health, the image of the selling company, and trust in the purchased products [42].

The undertaken study responded to the lack of analyses comparing the purchasing reasons in the V4 countries. Therefore, the presented research fills this gap, and the results showed differences in reasons for buying organic food between Polish, Czech, Hungarian, and Slovak consumers. They are only partly consistent with the research results presented above, and only looked at some countries of the V4 group. The study showed differences in the reasons for purchasing organic food between the studied countries. These differences may be due to many social and economic factors. The presented characteristics of consumer opinions may be helpful for companies and institutions that develop advertising and educational campaigns. The use of appropriate arguments can speak to consumers more effectively, cause more frequent purchases of organic products, and positively affect the environment [19].

The most important reasons for choosing organic food in all V4 countries were the absence of GMOs and chemicals and preservatives. In the literature, many authors emphasized the importance of these factors when choosing organic food, especially in the case of consumers from developed countries, such as France or Spain [30,43], but also developing countries [44]. Research results indicate incomplete knowledge of consumers, e.g., about the lack of allergens in organic products [45,46,47], while they have significant concerns about the use of modified food [48,49,50].

Some studies have found that the consumers’ perception of organic products depends on the frequency of organic consumption. Consumers of organic food have a more favorable impression of these products [51]. The reasons for choosing organic food most often mentioned in the literature indicate health aspects, i.e., not modifying food, not saturating it with chemicals and preservatives, and presenting overall high health values [35,52,53]. The obtained results relate, among others, to Czech consumers, which have been confirmed in national surveys [54]. They demonstrated that better flavor, health concerns, and food freshness were the most important attributes influencing the decision-making of the young Czechia consumer. The results relating to the declarations of Poles and Hungarians as to the degree of consumption of organic products contradict the research of Rodríguez-Bermúdez et al. [30]. The authors showed a relationship between the frequency of consumption of organic products and consumers’ opinions. This may be because the organic food market in Poland and Hungary, similarly to Serbia, is at the initial stage of development and can be considered developing [55].

The good taste of organic products is appreciated all over the world, both by consumers from Europe, e.g., from France, Spain, and from Asia, e.g., from Vietnam and Bangladesh [30,43,56,57]. A similar tendency occurred in three V4 countries, i.e., Czechia, Hungary, and Slovakia. Taste is a criterion often indicated by consumers when choosing and evaluating various food products [58,59,60]. The different opinion of Polish consumers may result from their idea about the high price of these products.

In addition to health, contribution to preserving the environment is often a highly ranked reason for purchasing organic food [53,56,61]. The surveyed V4 consumers hardly considered this issue. However, Yadav and Pathak (2016) confirmed environmental concerns in a study conducted among young consumers in the Czech Republic. On the other hand, the surveyed consumers of the V4 countries (including the Czechs) were not fully aware of the impact of food production on the environment. However, it must be stated that the fact that this reason was included in the created model proves its importance. The low position in the model may be influenced by the fact that environmental aspects are not always considered in the surveyed countries, both by consumers and food-producing plants [4,62].

This work also analyzes barriers to purchasing ecological products. Too high a price is the main barrier limiting the purchase of organic food. Such a reason for resignation from the purchase was indicated in previous studies in Poland [63]. Similar barriers were indicated by French and Spanish consumers [30,43], and also by Asian consumers [64,65]. However, Švecová and Odehnalová [54], in studies conducted among young Czech consumers, showed that young consumers are willing to pay far higher prices for good-quality food, and as a result the high price no longer acts as one of the main barriers in the purchase of organic food. Around 90% of respondents were willing to pay a higher price. In contrast, Spanish consumers would buy more organic food if the price was higher, but only 10–30% higher than conventional products [30].

The inconvenience associated with the purchase is also a significant barrier. These barriers function mainly in countries where the market of organic products is just developing, i.e., in Slovenia, Ireland, and in Asian countries [20,66,67]. This barrier also limits the purchase of organic food in the V4 countries, mainly in Poland and Hungary.

Other barriers to purchasing organic food identified in the literature include the lack of a habit of eating organic products and the lack of consumer confidence [30,68]. Organic food consumption is also restricted by organic label fraud [67]. Lack of trust was the least important factor for V4 consumers, especially from Poland. This may be due to the current belief in certifying authorities. This would indicate changes in this respect compared to previous studies [33,60], and a growing trust in producers or retailers that are able and willing to monitor their organic suppliers and ensure that organic standards are respected [69,70].

## 5. Limitation

The studies performed have a particular limitation. Firstly, they concern countries in central Europe with a similar level of development and a similar history. Therefore, applications may not apply to countries with different geographic and political circumstances. Another limitation is the volatility of consumer preferences and production trends. Research on organic food should be carried out systematically to observe the dynamics of its product development and interest in it by the inhabitants of the studied countries. The results of similar studies carried out in, for example, ten years, may indicate other reasons and barriers regarding the purchase of organic products.

## 6. Conclusions

Organic production performs two social functions: it produces goods by caring for and contributing to protecting the natural environment and rural development. On the other hand, it delivers organic products to the market, responding to reported consumer demand. In the European Union countries, including the V4 countries, more and more consumers report their need for food produced using natural substances and techniques. Organic food, however, still accounts for a small percentage of total agricultural production in the surveyed countries, although more and more farm areas are allocated to organic farming. Organic agricultural production expressed in hectares in the years 2012–2020 in Hungary increased on average by as much as 12.8% annually, in Slovakia by 2.7%, the Czech Republic by 1.9%, and in Poland, it decreased by 3.6%. A more significant increase in organic production will most likely take place when Regulation (EU) 2018/848 of the European Parliament and the Council of 30 May 2018 on organic production and labeling of organic products and repealing Council Regulation (EC) No 834/2007 is implemented in the food policies of each of the V4 countries.

To fully utilize the potential of the organic farming sector and organic aquaculture and to ensure their sustainable development, it is necessary to define the goals and activities to be implemented by the Minister of Agriculture in individual countries to produce organic food together with its promotion. When planning an organic food promotion strategy, the consumers’ motives when buying organic food should be considered. For example, in the V4 countries, the high health, nutritional value, and taste of organic products should be promoted. Information campaigns should be strengthened regarding the absence of allergens, GMO modifications, chemicals, and preservatives. The distribution process of such food should also be improved, for example, by developing a network of local stores and points of sale that will be promoted under a familiar brand, and consumer confidence in organic food should be increased, for example by organizing open days at organic farms or processing plants. Promotional campaigns should be common to all producers of organic food under the patronage of the Ministry of Agriculture. Such activities will contribute to the increase in demand for organic food in the studied countries. They will eliminate the differences in the consumption of this food between western and northern Europe and the countries of the V4 group.
The development of the organic product market will depend on the behavior and choices of consumers who perceive organic food as healthier, safer, and tastier;The research shows that respondents from Poland, the Czech Republic, Hungary, and Slovakia were guided by similar behaviors regarding the purchase of organic food. However, the attitudes of the respondents slightly differed between countries. Regarding the reasons for choosing organic food, the most important thing was that it is non-genetically modified food, especially for Polish consumers. The following were mentioned: lack of chemical compounds (Slovaks and Czechs), high health value of such food (Czechs and Slovaks), and excellent taste (Hungarians). The most critical barriers to purchasing are the price (Poles and Hungarians), difficult access (Poles and Hungarians), and the short expiry time of such products (Slovaks).To stimulate the consumption of organic food, it is necessary to take measures on the supply and demand side, which serve to diversify distribution channels and improve the availability of organic food and information on this type of product.The obtained results are helpful for organizations working with organic producers to improve the availability of organic products and to organize consumer education about the health and environmental benefits of consuming such products.

## Figures and Tables

**Figure 1 nutrients-13-04351-f001:**
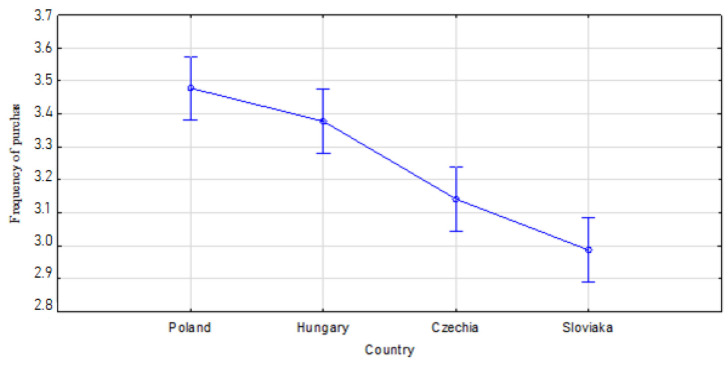
The frequency of purchasing organic food on a 1–5 point scale; where 1—very rarely, 5—very often; F (3.295) = 20.556; *p* < 0.001.

**Table 1 nutrients-13-04351-t001:** Characteristics of the research sample.

Age	Sex	Czechia		Hungary		Poland		Slovakia	
		N	%	N	%	N	%	N	%
from 20 to 34 years	female	80	10.84	82	11.11	92	12.47	92	12.47
	male	85	11.52	87	11.79	95	12.87	96	13.01
from 35 to 49 years	female	106	14.36	105	14.23	103	13.96	106	14.36
	male	112	15.18	107	14.50	105	14.23	111	15.04
from 50 to 65 years	female	87	11.79	94	12.74	94	12.74	94	12.74
	male	86	11.65	85	11.52	88	11.92	89	12.06
65 years or over	female	106	14.36	110	14.91	97	13.14	90	12.20
	male	76	10.30	68	9.21	64	8.67	60	8.13

**Table 2 nutrients-13-04351-t002:** Frequency of organic food purchase.

Country V4	Size Test	Very Rarely 1	Rarely 2	Moderately 3	Often 4	Very Often 5
		Data in %				
	N = 2952	Chi-Squared Test = 85.577; *p* < 0.001				
Czechia	738	11.11	25.47	21.82	21.27	20.33
Hungary	738	9.35	20.46	22.36	18.70	29.13
Poland	738	7.99	20.60	20.73	16.94	33.74
Slovakia	738	15.18	27.37	18.97	20.46	18.02

**Table 3 nutrients-13-04351-t003:** Reasons for purchasing organic food.

Factor	Model of Discriminant Analysis: λ Wilks: 0.326; F (33.865) = 87.987; *p* < 0.001						
	λ Wilks	F	*p*	Classification Functions			
				Czechia	Hungary	Poland	Slovakia
Tastes good	0.350	51.188	<0.001	2.278	2.453	1.379	2.290
Low in calories	0.332	11.648	<0.001	1.589	1.652	1.188	1.500
Little processed	0.331	10.583	<0.001	1.535	1.418	1.260	1.711
Lots of nutritional value	0.361	80.337	<0.001	2.068	2.012	1.067	2.329
High health benefits	0.383	130.386	<0.001	2.895	1.772	1.692	2.800
Environmentally friendly	0.338	27.018	<0.001	0.569	0.735	0.194	0.813
No preservatives	0.339	29.684	<0.001	1.096	1.212	1.596	1.009
Not genetically modified	0.333	15.060	<0.001	5.413	5.365	5.879	5.454
Does not cause allergies	0.329	6.485	<0.001	0.772	0.710	0.731	0.529
No chemicals	0.340	15.557	<0.001	2.596	2.436	2.295	2.687
Constant				−46.139	−42.279	−33.817	−48.081

**Table 4 nutrients-13-04351-t004:** Barriers to purchasing organic food depending on the country.

Factor	Model of Discriminant Analysis: λ Wilks: 0.503; F (12.779) = 192.78; *p* < 0.001						
	λ Wilks	F	*p*	Classification Functions			
				Czechia	Hungary	Poland	Slovakia
Too expensive	0.558	108.212	<0.001	2.333	3.520	3.799	2.824
Hardly available	0.542	76.633	<0.001	2.294	3.208	3.583	2.600
No trust in organic food	0.521	34.971	<0.001	1.357	1.201	1.025	1.651
Short expiry time	0.534	60.540	<0.001	2.216	2.154	1.699	2.570
Constant				−12.888	−18.873	−19.737	−17.301

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
