# Peer review of "Organic Food in the Diet of Residents of the Visegrad Group (V4) Countries—Reasons for and Barriers to Its Purchasing"

_nutrients, 2021, doi:10.3390/nu13124351_

Round 1
Reviewer 1 Report
This manuscript presents a basic study of the factors that influence purchase behaviour of organic food in V4 countries. Authors asked consumers about the main reasons and barriers to buy organic food.
Although there are a plethora of studies about organic food in the consumer behaviour field conducted in Western Europe, there has not been enough writing about this type of food in Central Europe.
In general, the manuscript presents a good structure and it is quite easy to follow the ideas of the authors. However, I recommend to authors a minor revision of the manuscript before publishing it.
Materials and Methods
Table 1. Regarding this table, I suggest the authors move it to the appendix or supplementary material. Regarding to this, the description of the measures used in the study should be presented instead of table 1. Moreover, authors should check the formatting table.
Sentences of line 113 to 114 are irrelevant and I suggest dropped out to the manuscript.
Lines 119-125. It is not necessary to present the formula used to determine the sample size. However, it is important to indicate the confidence level and the standard error considered to calculate the sample size.
Lines 126-127. Authors said that the sample size for each country should be 738 respondents. This is similar to the distribution presented in table 2. But, in lines 131- 134 authors said the sample was 185-184 for each country. Please check this.
Table 2. Authors did not describe in the main text the characteristics of the sample. The table shows an overrepresentation of females of 65 years old and older in the four countries. Is this representative of the population of each country? I suggest including the description of the sample’s characteristics and if this is representative of the population of each country.
Results
I encourage the authors to improve the presentation of the results of table 3 (lines 163- 172). I suggest emphasise on similarity or differences among countries and highlighting lower and higher frequency of food purchase.
Lines 199 to 200. The sentence “A vital reason…” is repetitive. I suggest dropped out to the manuscript.
Discussion
Line 137. References are missing.
Lines 236 to 248. In the Introduction section authors mention to the highest consumers of organic food. But the results are compared to countries with less consumption of this kind of food. I suggest including some data of countries with higher consumption of organic food (e.g. Denmark, Switzerland).
Lines 249-260. Authors said that “The resulting difficulties as to the correct interpretation are influenced by the small number of studies dealing with the issues raised by the authors, which indicates the sense of conducting systematic and repeated research”. I disagree with the authors. There are several studies that analyze the impact of knowledge about organic production on purchase behavior. In a fast search I found this one which was conducted in Poland.
Wojciechowska-Solis, J.; Barska, A. Exploring the Preferences of Consumers’ Organic Products in Aspects of Sustainable Consumption: The Case of the Polish Consumer. Agriculture 2021, 11, 138. https://doi.org/10.3390/agriculture11020138
Minor issues such as differences in font size of several paragrah should be corrected.
Author Response
We sincerely thank you for all your comments. In Attachment are the responses to the review.
Reviewer 2 Report
Referee Report on Manuscripts ID: Nutrients-1418014
“Organic food in the diet of residents of the Visegrad Group 2 (V4) countries – reasons and barriers to its purchasing”
The main objective of this study is to identify consumers’ purchasing behavior in the context of organic food products. It is a well-written paper. However, I do have some minor concerns:
- How do you recruit participants for the survey? Please mention the recruitment process.
- Did you pay participants? What was the average survey length?
- Was it an online survey? Did the authors hire a survey company to conduct the survey or use an online platform? Or it is a face-to-face survey?
- Please provide additional details regarding survey participant consent. I would recommend adding this information to the methodology section. Please ensure that you have specified (1) whether consent was informed and (2) what type you obtained (for instance, written or verbal, and if verbal, how it was documented and witnessed). If the need for consent was waived by the IRB, please include this information.
- L 278-283 and L 368-383: fix the font size.
Author Response

(The authors gave the same response as above.)
